# Immunologic Alteration After Total En-Bloc Spondylectomy with Anterior Spinal Column Reconstruction with Frozen Tumor-Containing Bone Autologous Grafts: A Case Report in a Prospective Study

**DOI:** 10.3390/curroncol32080432

**Published:** 2025-07-31

**Authors:** Hisaki Aiba, Hiroaki Kimura, Ryu Terauchi, Nobuyuki Suzuki, Kenji Kato, Kiyoshi Yagi, Makoto Yamaguchi, Kiyoka Murakami, Shogo Suenaga, Toshiharu Shirai, Ayano Aso, Costantino Errani, Hideki Murakami

**Affiliations:** 1Department of Orthopedic Surgery, Graduate School of Medical Sciences, Nagoya City University, Nagoya 467-8601, Japan; hiroaki030301@yahoo.co.jp (H.K.); nobuyuki.suzuki@me.com (N.S.); kenkenkatoken@yahoo.co.jp (K.K.); kiyoshi.yg@gmail.com (K.Y.); yamaguchi1220makoto@gmail.com (M.Y.); s.y.h.h.k.r@gmail.com (S.S.); hmuraka@med.nagoya-cu.ac.jp (H.M.); 2Department of Orthopaedics, Graduate School of Medical Science, Kyoto Prefectural University of Medicine, 465 Kajii-cho, Kawaramachi-Hirokoji, Kamigyo-ku, Kyoto 602-8566, Japan; ryu@koto.kpu-m.ac.jp (R.T.); shirai.t77@gmail.com (T.S.); 3Department of Orthopaedics, Toyohashi Medical Center, 50 Hamamichiue, Iimura-cho, Toyohashi 444-0836, Japan; kiyozo.dx.2525@icloud.com; 4Clinica Ortopedica e Traumatologica III a Prevalente Indirizzo Oncologico, IRCCS Istituto Ortopedico Rizzoli, Via Pupilli 1, 40136 Bologna, Italy; ayano.aso@studio.unibo.it (A.A.); costantino.errani@ior.it (C.E.)

**Keywords:** cryotherapy, bone metastases, surgery, total en-bloc spondylectomy, frozen autologous bone graft, immune activation, cryoimmunity, immunosequencing

## Abstract

Cryotherapy could stimulate immune responses, including abscopal effects. This study explored a novel approach for treating spinal bone tumors, utilizing frozen tumor-containing autologous bone grafts for anterior spinal reconstruction after total en-bloc spondylectomy. T-cell receptor (TCR) repertoire changes were analyzed pre- and post-surgery to assess immune activation, focusing on T-cell diversity. Blood samples were collected pre- and post-surgery, followed by RNA extraction and immunosequencing. Results showed a decrease in the diversity and abundance of CDR3 regions in TCR α and β chains postoperatively, indicating the emergence of selective T-cell clones that may influence immune responses. This suggests that transplanting frozen tumor-containing autologous bone grafts impacts the immune system, potentially enhancing antitumor immunity. This approach offers promising insights for developing treatments aimed at activating immune responses and improving outcomes for spinal bone tumors.

## 1. Introduction

Activation of the immune response following cryotherapy was first described by Shulman et al. in the 1960s; they observed that the transplantation of autologous frozen tumors resulted in the release of large amounts of antigens, triggering an immune reaction [1]. Moreover, Soanes et al. reported the regression of distant metastases in patients with prostate cancer treated with cryoablation at the primary site [2]. These phenomena are related to structural changes in proteins after freezing, resulting in the exposure of more immunogenic epitopes and enhancement of their immunogenicity [3]. Although immune tolerance, which can be exploited by tumor cells to escape from the host immune system, is essential for preventing hyper-reactions to self-antigens, cryotherapy may provide a means to elicit a more robust immune response against tumors [3].

The abscopal effect is a phenomenon in which radiotherapy causes regression of the targeted tumor and shrinks tumors at metastatic sites [4]. This effect occurs through the activation of systemic antitumor immune responses triggered by radiation [4]. The abscopal effect has been reported in several malignant tumors, such as malignant melanoma and non-small cell lung cancer [5,6,7], with various types of local treatment, including cryotherapy [8,9].

Cryotherapy for bone tumors is initiated as a reconstruction method for bone tumors of the extremities [10,11]. It has been experimentally proved that tumor cells in the tumor-bearing bone are completely killed by immersion in liquid nitrogen for 20 min [12]. Currently, autografts of frozen bone are considered safe and effective and are widely utilized [10,11]. This technique was utilized for spinal metastatic or primary bone tumors; after total en-bloc spondylectomy (TES), anterior spinal column reconstruction is performed with frozen tumor-containing bone autografts (second-generation TES) with the intention to activate cryo-immunity [12,13] (Figure 1). A case series analysis of patients who received second-generation TES reported that among 51 patients, 24 (47.1%) had no growth or emergence of metastatic lesions [14]. Additionally, four patients, two each with breast cancer and thyroid cancer, experienced spontaneous regression of metastatic lesions or a decrease in tumor markers without chemotherapy [14].

This prospective study investigated immune activation following second-generation TES, focusing on changes in the T-cell receptor (TCR) repertoire after surgery. In this case report, the methodology was described, and a representative case was presented as a preliminary result of an ongoing prospective study.

## 2. Materials and Methods

### 2.1. Patient

A 10-year-old boy was diagnosed with conventional osteosarcoma of the right distal femur. The patient underwent perioperative chemotherapy with tumor resection. Three years later, the patient underwent above-knee amputation due to recurrence at the surgical site. Seven months later, a solitary metastasis in the 12th thoracic spine was pointed at 14 years old. To achieve a disease-free status and prevent possible invasion of the spinal canal, second-generation TES was planned (Figure 2).

### 2.2. Tumor Resection and Reconstruction

Second-generation TES was performed with the patient in the prone position under general anesthesia. A midline incision was made to expose the posterior surface of the thoracic vertebrae. The bases of the 12th ribs were cut bilaterally. The spinous process and inferior articular facet of the 11th thoracic vertebra were resected. Surgical thread wire saws were inserted from both sides of the 12th thoracic vertebral foramen, and the vertebral arch and transverse processes of the 12th thoracic vertebra were extracted. The 12th thoracic nerve roots were resected bilaterally. The anterior surface of the 12th thoracic vertebra was gently separated from the surrounding tissues. After inserting pedicle screws and setting-rods to stabilize the vertebral column (covering the 10–11th thoracic and 1–2nd lumbar vertebrae), the 12th thoracic vertebra was resected en bloc. The tumor-containing 12th vertebral body was crushed into small pieces, frozen with liquid nitrogen (−196 °C for 20 min), then packed into a cage with autologous, unfrozen, crashed bones (harvested from the bases of the 12th ribs, as well as the spinous process and facet of the 11th thoracic vertebra) for reconstruction of the 12th vertebral body. The total operative time was 322 min, and the total blood loss was 311 mL. The bony fusion of the anterior column was confirmed three months postoperatively.

### 2.3. RNA Extraction and the Assessment of Integrity for the Sequence

RNA was extracted from whole blood using the PAXgene Blood RNA System (Becton Dickinson, Franklin Lakes, NJ, USA) just before second-generation TES and 3 months postoperatively. The concentration and quality were tested using a NanoDrop 2000 spectrophotometer (Thermo Fisher Scientific, Waltham, MA, USA). Agilent 2100 (Agilent Technologies, Santa Clara, CA, USA) was used for the precise evaluation of RNA integrity, which was assessed based on the RNA Integrity Number value and flatness of the electropherogram baseline.

### 2.4. Immunome Sequencing Method

A 5′ rapid amplification of cDNA ends was performed for reverse transcription. The reverse transcription products were amplified using PCR and purified using DNA magnetic beads. The terminal repair was performed using End Prep Enzyme Mix (QIAGEN, Venro, The Netherlands) with sequencing adapters at both ends. Subsequently, clean DNA beads were purified and amplified using the P5 and P7 primers. Bead-based methods were used to purify the PCR products, which were then evaluated for quality with the Qsep100 system (Bioptic, Taiwan, China). Quantification was performed using the Qubit 3.0 Fluorometer (Invitrogen, Carlsbad, CA, USA). Multiple libraries with unique indices were combined (multiplexed) and subsequently loaded onto an Illumina MiSeq or NovaSeq device following the instructions provided by the manufacturer (Illumina, San Diego, CA, USA). Sequencing was conducted using a 2 × 250 or 2 × 300 paired-end configuration, and image analysis and base calling were conducted using the MiSeq Control Software (Version 4.0)/Novaseq Control Software (Version 1.8) on the MiSeq/Novaseq instrument (Illumina).

### 2.5. Data Analysis

The raw FASTQ files underwent an initial quality evaluation (Table A1). Poor-quality bases and adapter sequences (Q-value < 20) were trimmed using Cutadapt (version 1.9.1) to produce cleaned, high-quality data. Paired-end reads were then merged using FLASH (version 2.2.00). The resulting merged sequences were blasted against the IMGT reference database to determine the best correspondence between germline V(D)J gene segments and the sequences within the Complementarity-Determining Region 1 (CDR1), CDR2, and CDR3. V(D)J recombination is a distinctive genetic rearrangement process that occurs exclusively in developing lymphocytes during early T- and B-cell development [15,16]. This mechanism results in many V(D)J combinations, engineering a highly diverse repertoire of immunoglobulins and T-cell receptor combinations. After obtaining the clone and CDR sequence information, customized scripts were written to perform the following statistical analyses: For diversity analysis, CDR3, an important region for antigen recognition [17], resulting from V(D)J recombination, was analyzed using rank–abundance curves. Additionally, the richness [18] of CDR3 types was analyzed using the abundance-based coverage estimator (ACE) [19], Chao1 [19], Shannon [20], and Simpson indices [21].

## 3. Results

### 3.1. VJ Gene Recombinations

The merged sequences were BLASTed against the IMGT reference database to identify the best match for the germline *VJ* genes. The abundance of clones after mapping and clustering was calculated for each chain type at various time points. The Circo plots illustrate the interaction between *TRAV–TRAJ* or *TRBV–TRBJ* gene segments, where the connecting lines represent the pairing frequency of specific segments, with thicker lines indicating more common combinations (Figure 3a–d). The heatmaps revealed the dominant pairing genes between *TRAV–TRAJ* or *TRBV–TRBJ*, indicating clonal expansion and a favored V-J gene combination in the dataset (Figure 3e–h). Analysis of the full-length sequence and abundance of the identified VJ region in each sample is shown in Appendix A.

### 3.2. Diversity Analysis

The rank–abundance curves showed that the diversity of the CDR3 α chain in the preoperative period exhibited a higher level of repertoire diversity and richness compared to that during the postoperative period (Figure 4a). Specifically, the preoperative CDR3 α chain had approximately 2967 unique clonotypes, whereas the postoperative sample had approximately 1865. The ACE and Chao1 indices, which estimate total unseen diversity, were also higher in the preoperative CDR3 α chain. The Shannon index was 11.085 for the preoperative CDR3 α chain, slightly exceeding the postoperative value, indicating a more balanced distribution of clones. Both samples demonstrated high Simpson indices (close to 1), implying that overall diversity remained high. Likewise, the diversity of the CDR3 β chain in the preoperative period exhibited a higher level of immune repertoire diversity compared to that during the postoperative period (Figure 4b). The preoperative CDR3 β chain had approximately 5070 unique clonotypes, higher than the 4050 clonotypes in the postoperative sample. The ACE and Chao1 indices were also higher in the preoperative CDR3 β chain, suggesting that the preoperative repertoire was more extensive and included many rare clones. The Shannon index for the preoperative period (11.974) was slightly higher than that of the postoperative period (11.630), indicating a more even distribution of clones in the preoperative period. Both samples showed high Simpson indices (close to 1), indicating that diversity was high and no single clone dominated either repertoire (Table 1).

### 3.3. Patient’s Outcome

The patient remained disease-free for 1 year following second-generation TES; however, a solitary metastasis was noted in the right lower lobe. The patient received a lobectomy and is currently disease-free (2 years after second-generation TES).

## 4. Discussion

This case report presents preliminary results from a prospective study of immunologic alterations following second-generation TES, focusing on the repertoire of TCRs. The findings demonstrated that the repertoires of the CDR3 regions of the TCR α and β chains decreased after surgery, suggesting that more selective clones may have emerged and influenced immune responses.

TCRs are transmembrane glycoprotein heterodimers mainly formed by the combination of α and β chains [22]. The genes encoding the α (*TRA*) and β (*TRB*) chains consist of multiple non-continuous segments, including variable (V), diversity (D) segments, and joining (J) segments [22], and TCRs acquire their diversity through V(D)J recombination [22]. Diversity in humans is approximately 2 × 10^7^ different clonotypes [23]. With exposure to various antigens, the diversity of the TCR repertoire dynamically changes throughout its life but it generally decreases with aging [24].

The tumor microenvironment (TME) contains numerous immune-related cells, and cytotoxic T cells play a central role [25,26]. In the TME, tumor antigens are presented by antigen-presenting cells, resulting in the clonal expansion of T cells and restriction of the TCR repertoire [27]. The relationship between TCR repertoire and prognosis has been studied in various tumors [28,29]. Campana et al. analyzed TCR repertoires based on colorectal cancer datasets from the Cancer Genome Atlas and found that patients with numerous TCR repertoires tended to have longer overall survival compared to those with fewer TCR repertoires [30]. Moreover, based on an analysis of patients with metastatic breast cancer, lymphopenia before chemotherapy and lower TCR diversity were independent predictors of prognosis [31]. Thus, the loss of TCR diversity is believed to result from the aggressiveness of tumors and may lead to immune system failure [22].

Tumor mutation burden (TMB) influences the TCR repertoire, as higher values often lead to increased tumor clonality [22]. Tumors with higher TMB levels produce more tumor neoantigens with high affinity for the human leukocyte antigen, thereby increasing their chances of being recognized by T cells [22]. Notably, a study suggested that patients with higher TMB and clonality respond better to immunotherapies, especially in tumors with mismatch repair deficiency (MMR-d) or high microsatellite instability (MSI), due to increased tumor neoantigen production, thereby promoting tumor-specific lymphocyte infiltration and activation [32]. Additionally, in colorectal cancer, MSI-high/MMR-d tumors show higher T-cell infiltration and more clonally expanded TCR repertoires that proliferate further after immunotherapy [33].

Matzinger et al. reported that when tumor tissue is frozen, necrosis primarily occurs in the center of the tumor. In contrast, apoptosis mainly occurs in the periphery of the tumor, depending on the freezing temperature [34]. In necrotic tissue, various molecules, including DNA, RNA, and damage-associated molecular patterns (DAMPs) are released from ruptured cells as danger signals [34]. High mobility group box protein 1 (HMGB1), heat shock proteins, calreticulin, and GP96 have been identified as DAMPs, and HMGB1 is considered a key factor in activating CD8+ T cells through the innate immune system via Toll-like receptor-4 [35,36]. Furthermore, necrotic tissue creates a cytokine-rich environment—such as interleukin (IL)-2, interferon (IFN)-γ, tumor necrosis factor-α, and IL-12—that induces CD8+ T-cell responses, and is presumably in a state conducive to immune reactions against the antigens released from frozen tissue [37]. The TME surrounding the transplanted frozen tissue enhances T-cell activation by tumor neoantigens, with epitopes altered by freezing, potentially resulting in the clonal proliferation of T cells. The results of the current case report indicate a decrease in repertoire diversity after second-generation TES, which may have been caused by reactions to frozen tissue transplantation.

Immune checkpoint inhibitors (ICIs) suggestively enhance the abscopal effect [38]. According to a case report on patients with melanoma treated with cytotoxic T-lymphocyte antigen-4 inhibitors, the radiotherapy of thoracic spinal lesions reduced other metastatic tumors and increased NY-ESO-1 antibodies, a key factor in mediating the abscopal effect, was observed during radiotherapy [38]. Analysis of the TCR repertoire is critical to predicting the efficacy of ICIs [39]. Analysis of blood samples from patients with melanoma who received nivolumab revealed increased TCR clonality after ICI treatment was observed in both peripheral and tumor tissues [39]. Similarly, a randomized controlled study of neoadjuvant ipilimumab and nivolumab for early-stage colon cancer showed that ICI responders had higher baseline CD8+PD-1+ T-cell infiltration and TCR clonality [40]. Similarly to radiotherapy, cryotherapy destroyed tumor tissue and increased tumor antigens potentially leading to a synergistic immune response, especially when combined with ICI [41]. Thus, frozen tissue transplantation used in second-generation TES may enhance antitumor effects when combined with ICIs.

As the results are based on a single case report, it is difficult to gain a comprehensive understanding of the changes in TCR repertoires. Additionally, control data from patients who did not undergo frozen tissue transplantation are necessary for a meaningful comparison. Furthermore, the effects of cryoimmunization may vary depending on factors such as the type of primary lesion, amount of transplanted tissue, and subsequent treatments. To the best of our knowledge, this study is the first to employ TCR repertoire analysis following second-generation TES. Further validation and verification studies are planned.

## 5. Conclusions

Through TCR repertoire analysis, this study demonstrated that the transplantation of frozen tumor-containing autologous bone grafts has an impact on the immune system. 

## Figures and Tables

**Figure 1 curroncol-32-00432-f001:**
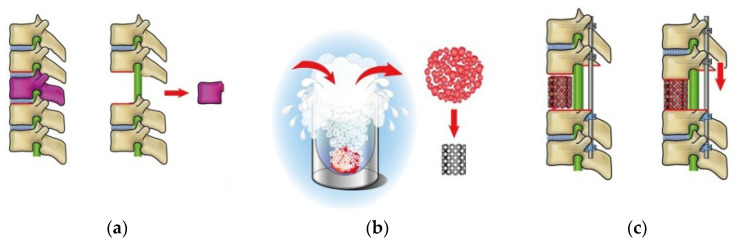
Schema of second-generation TES. (**a**) en-bloc excision of the vertebra (or vertebrae); (**b**) the resected lamina and vertebral body, containing tumors, are used as bone grafts for anterior spinal reconstruction. These are placed in liquid nitrogen (−196 °C) for 20 min before being crushed and packed into a titanium cage with a sufficient amount of autologous, unfrozen grafts harvested during the resection procedure); (**c**) the cage is placed between the adjacent healthy vertebral bodies, and spinal shortening is performed to stabilize the cage [12].

**Figure 2 curroncol-32-00432-f002:**
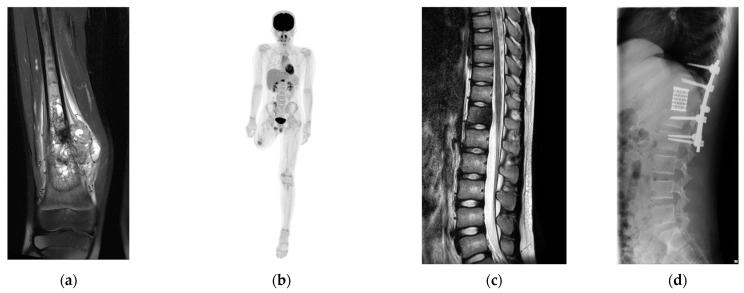
Clinical course of a 10-year-old male with conventional osteosarcoma. (**a**) Image of the primary lesion (right proximal femur, short τ inversion recovery). (**b**) Positron emission tomography shows a single metastasis in the 12th thoracic spine. (**c**) Sagittal image of the spine (T2-weighted image) and (**d**) after surgical resection of the 12th thoracic spine and reconstruction.

**Figure 3 curroncol-32-00432-f003:**
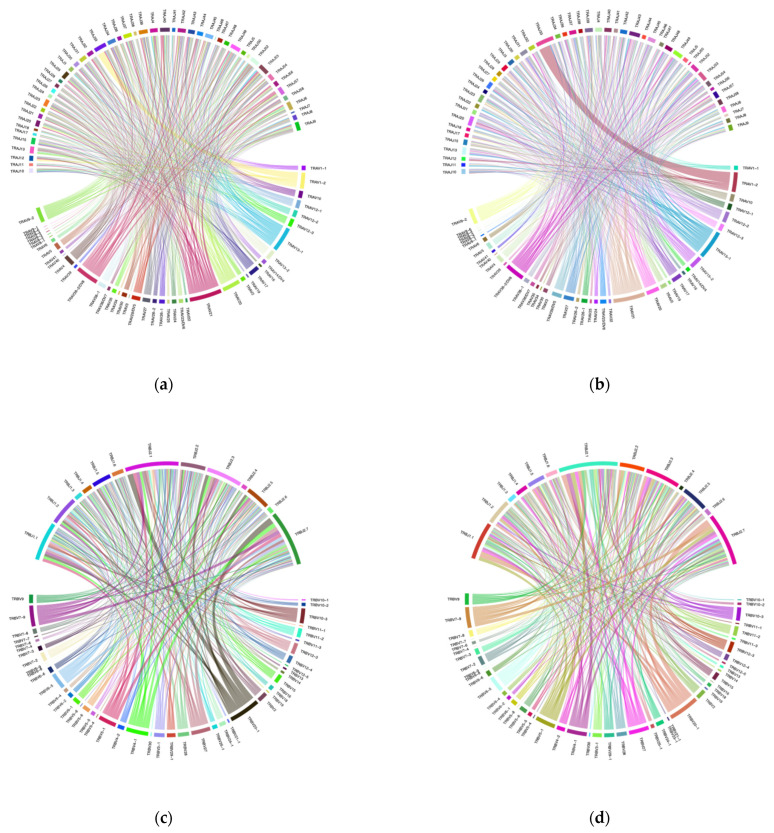
VJ genes recombination. (**a**) Circo plots for the combination of *TRAV–TRAJ* genes before surgery; (**b**) Circo plots for the combination of *TRAV–TRAJ* genes 3 months after surgery; (**c**) Circo plots for the combination of *TRBV–TRBJ* genes before surgery; (**d**) Circo plots for the combination of *TRBV–TRBJ* genes 3-month after surgery; (**e**) heatmap for the combination of *TRAV–TRAJ* genes before surgery; (**f**) heatmap for the combination of *TRAV–TRAJ* genes 3-month after surgery; (**g**) heatmap for the combination of *TRBV–TRBJ* genes before surgery; and (**h**) heatmap for the combination of *TRBV–TRBJ* genes 3-month after surgery.

**Figure 4 curroncol-32-00432-f004:**
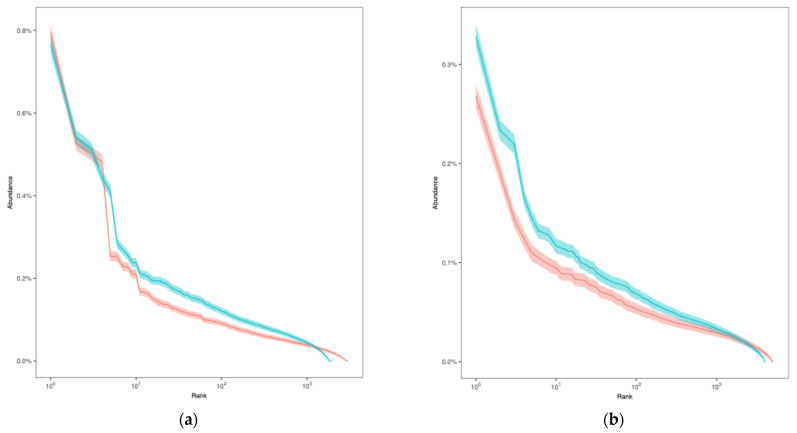
Rank–abundance curve plot. The horizontal axis represents CDR3 abundance ranking. The vertical axis represents the CDR3 abundance frequency corresponding to the number, and the shaded area indicates the 0.95 confidence interval obtained using the bootstrap method. The rank–abundance curves indicated a distribution pattern characterized by a few highly abundant clones at the top ranks and a long tail of numerous rare sequences. (**a**) CDR3 α chain; (**b**) CDR3 β chain. The blue tones indicate “preoperative” and red tones indicate “postoperative”.

**Table 1 curroncol-32-00432-t001:** Diversity in the CDR3 chains. ACE, abundance-based coverage estimator; CDR, complementarity-determining region.

Analyzed Points	Richness	ACE	Chao1	Shannon	Simpson
CDR3 α chain (preoperative)	2967	3017	3006	11.09	0.9993
CDR3 α chain (postoperative)	1865	1921	1940	10.46	0.9990
CDR3 β chain (preoperative)	5070	5154	5174	11.97	0.9997
CDR3 β chain (postoperative)	4050	4094	4101	11.63	0.9996

## Data Availability

We provide details regarding where data supporting reported results in a Appendix A.

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
