# Peer review of "Immunologic Alteration After Total En-Bloc Spondylectomy with Anterior Spinal Column Reconstruction with Frozen Tumor-Containing Bone Autologous Grafts: A Case Report in a Prospective Study"

_curroncol, 2025, doi:10.3390/curroncol32080432_

Round 1
Reviewer 1 Report
Comments and Suggestions for Authors The case report submitted introduces an innovative surgical and immunological approach for the treatment of spinal bone tumours. The authors describe the use of frozen, tumour-bearing autologous bone grafts for anterior column reconstruction following total en-bloc spondylectomy (TES). The central hypothesis is that this approach may not only provide structural stability but also trigger systemic immune responses, potentially through mechanisms akin to the abscopal effect. The study design focuses on a single clinical case involving a 14-year-old patient with spinal metastasis secondary to osteosarcoma. The results section documents a postoperative reduction in TCR diversity, which was evident in both the α and β chains. While causality cannot be inferred from a single observation, these findings are consistent with the hypothesis that cryopreserved tumour-bearing bone may influence systemic T-cell dynamics.The manuscript avoids overinterpretation and acknowledges the need for larger studies to confirm these preliminary findings. Nonetheless, several limitations must be addressed. The study is based on a single case, which significantly limits its external validity. The postoperative analysis was limited to a single time point, and longitudinal data would be necessary to determine the persistence or evolution of the observed T-cell changes.
The strengths of the study lie in its clear formulation of a novel hypothesis, its robust methodological design, and its transparent discussion of the study’s limitations.
Author Response
Thank you for your thoughtful review. We appreciate your careful interpretation of the preliminary findings. We agree that the single-case design limits the broad applicability of the results, and we are committed to conducting larger, longitudinal studies to further extend the research. Your insights are valuable in guiding future research efforts, and we will ensure to address these limitations more thoroughly in subsequent work.
Reviewer 2 Report
Comments and Suggestions for Authors The case report submitted presents preliminary results from a prospective study of immunologic alterations following TES with anterior spinal column reconstruction with frozen tumor-containing bone, focusing on the repertoire of TCRs. The topic is original. To this day, there is no reported comprehensive analysis on the immunologic alterations following second-generation TES yet. Although the results are based on a single case report and the comparison control data from patients who did not undergo frozen tumor-containing bone transplantation are necessary for a more meaningful study, this report addresses a specific gap in the field. The conclusions are consistent with the evidence and arguments presented and the references are appropriate. My concern is about the reconstruction. How to garantee the bone fusion after the reconstruction with frozen tumor-containing bone?Author Response
Thank you very much for taking the time to review this manuscript. Please find the detailed responses below and the corresponding revisions in the re-submitted files. Comments 1: Cryotherapy could stimulate immune responses and induce abscopal effects. A novel technique was developed for treating spinal bone tumors involving the use of frozen tumor-containing autologous bone grafts for anterior spinal reconstruction following total en-bloc spondylectomy, with the aim of activating cryoimmunity. This study focused on analyzing changes in the T-cell receptor (TCR) repertoire after surgery to evaluate T-cell diversity. Blood samples were collected pre- and post-operatively, with subsequent RNA extraction and immunosequencing. Compared to pre-surgery samples, the diversity and abundance of the Complementarity Determining Region 3 regions of the TCR α and β chains decreased, suggesting that more selective clones may have emerged and influenced immune responses. Through TCR repertoire analysis, this study demonstrated that transplantation of frozen tumor-containing autologous bone impacted the immune system. This study is expected to provide a foundation for developing treatments that may enhance immune activation.
Response 1: Thank you for pointing this out. We agree with this comment.
We apologize for the previous lack of clarity. Regarding the reconstruction of the anterior column, autologous (unfrozen) bone is also transplanted. Accordingly, no clinical delays in bone union have been observed, and no related complications have been reported. Therefore, we have added the description “These are placed in liquid nitrogen (–196℃) for 20 min before being crushed and packed into a titanium cage with a sufficient amount of autologous, unfrozen grafts harvested during the resection procedure); “ in Figure legend 1.
In 2.2. Tumor resection and reconstruction (line 109-115). “The tumor-containing 12th vertebral body was crushed into small pieces, frozen with liquid nitrogen (−196°C for 20 min), then packed into a cage with autologous, unfrozen, crashed bones (harvested from the bases of the 12th ribs, as well as the spinous process and facet of the 11th thoracic vertebra) for reconstruction of the 12th vertebral body. The total operative time was 322 min, and the total blood loss was 311 mL. The bony fusion of the anterior column was confirmed three months postoperatively.”